# The bandwidth of perceptual awareness is constrained by specific high-level visual features

## Abstract

When observers glance at a natural scene, which aspects of that scene ultimately reach perceptual awareness? To answer this question, we showed observers images of scenes that had been altered in numerous ways in the periphery (e.g., scrambling, rotating, filtering, etc.) and measured how often these different alterations were noticed in an inattentional blindness paradigm. Then, we screened a wide range of deep convolutional neural network architectures and asked which layers and features best predict the rates at which observers noticed these alterations. We found that features in the higher (but not earlier) layers predicted how often observers noticed different alterations with extremely high accuracy (at the estimated noise ceiling). Surprisingly, the model prediction accuracy was driven by a very small fraction of features that were both necessary and sufficient to predict the observed behavior, which we could easily visualize. Together these results indicate that human perceptual awareness is limited by high-level visual features that we can estimate using computational methods.

## 1 Introduction

How much information do humans perceive when looking at a natural scene? Is our experience of the world rich and detailed (Lamme (2003); Block (2011), or is it sparse and limited ( Dehaene & Changeux (2011); Cohen et al. (2012))? What aspects of the visual world are observers aware of at any given moment? To try and answer these questions, researchers use paradigms like change and inattentional blindness to examine the limits of perceptual experience ( Jensen et al. (2011). In typical versions of these experiments, individual items change or appear in some unexpected manner and researchers measure how often observers notice these events.

However, there are limits as to what can be gleaned about perceptual experience using this approach for two main reasons. **First**, changes in these experiments typically involve alterations that are confined to individual objects/people within complex scenes: a shadow that appears and disappears ( Rensink et al. (1997)), a rail in the background that moves up and down ( O'Regan et al. (1999)), an individual in a gorilla costume walking amongst a group of people ( Simons & Chabris (1999)), etc. Therefore, it is difficult to extrapolate from these findings to broad generalizations about the overall bandwidth of perceptual experience. **Second**, stimuli in these paradigms change along many dimensions, making it difficult to synthesize them into a coherent whole. For example, the critical manipulations in these experiments involve a wide array of stimuli ranging from lower-level items like colors and simple shapes ( Mack & Rock (1997); Most et al. (2005)), to complex objects ( Simons et al. (2000)) or even entire scenes ( Cohen et al. (2011)). Therefore, creating general principles about perceptual experience from this diverse set of studies is difficult since the manipulated variables fall along numerous different perceptual dimensions.

Submitted to 4th Workshop on Shared Visual Representations in Human and Machine Visual Intelligence (SVRHM) at NeurIPS 2022. Do not distribute.

Here, we propose an algorithmic approach to examining the limits of perceptual awareness using computational models. We started by altering images of natural scenes in numerous ways and quantified how often observers noticed those alterations in an inattentional blindness blindness paradigm using Amazon's Mechanical Turk (N=1,260 observers). Finally, we sought to unify these behavioral results by building deep convolutional neural networks (dCNNs) based computational models that could predict the behavioral inattentional blindness rates. The idea behind this approach is that by building models that predict observers' behavior, we could then probe the specific internal features of these computational models to infer the critical features that best predicted human behavior.

## 2 Methods

### 2.1 Inattentional blindness behavioral paradigm

**Note:** All of the methods and analyses in this study were pre-registered to remove all experimenter bias (`osf.io/zr3ed`). **Participants**: 1,260 participants were recruited on Amazon's Mechanical Turk. Every subject gave informed consent. All procedures were approved by the MIT Institutional Review Board and the Committee on the Use of Humans as Experimental Subjects.

Overall, we created 21 experimental conditions with each condition corresponding to a different way of altering the periphery (Figure 1a). Participants were unaware of the experiment's true nature and were instructed to perform a simple face detection task at fixation. On each trial, participants were shown 7-30 images of natural scenes and reported whether the last image in the stream contained a face in the middle of it (Figure 1b). Each image was shown for 288ms, which approximately corresponds to the duration of one fixation in naturalistic viewing conditions ( Rayner (1998); Henderson (2003)). For the first 10 trials, half of the trials had a face target present at the end and half did not. At the end of each trial, a screen appeared that prompted the observer to say whether or not the last image had a human face in the middle.

On the critical trial, the last image in the stream was a target stimulus (Figure 1a). As soon as the critical stimulus disappeared, rather than be asked about if a face was in the middle, observers were instead asked another series of questions. Specifically: 1) "Did you notice anything different about that last trial?" 2) "If we were to tell you there was something different about that last trial, could you say what it was?" 3) "If we were to tell you there was something different about the very last image on that last trial, could you say what it was?" Only those participants who responded "no" to all of these questions were classified as having been inattentionally blind. If an observer responded "yes" to any of these questions, they were classified as having noticed the alterations.

## 3 Results

### 3.1 Inattentional blindness behavioral results

The results from these behavioral experiments are plotted in Figure 1c. Overall, there is substantial variance in the inattentional blindness rates between conditions. For example, virtually no observers noticed when the periphery was altered in the medium "metamer" conditions (92.5% inattentional blindness rate), while many observers noticed when the periphery was abstract and desaturated ( 35% inattentional blindness rate). The participants were also highly consistent in their responses (Spearman-Brown corrected, split-half reliability (r=0.82, P<0.00001). However, before attempting to model these results, we examined the reliability of this data by directly comparing the inattentional blindness rates of a subset of the conditions when using MTurk to those obtained when testing those exact conditions in a laboratory setting. Specifically, we took 6 conditions from our prior study that used the same experimental procedures and compared the behavioral results with the data obtained in the present MTurk study ( Cohen et al. (2021); (1) "Metamers" (small), 2) "Metamers" (large), 3) Texture-synthesis, 4) Inconsistent periphery, 5) Abstract periphery, and 6) Grey periphery) . The correlation between the laboratory and MTurk was remarkably high (r=0.98, P<0.0001). The fact that the laboratory and MTurk data is almost perfectly correlated is critical, as it implies that our methods for examining inattentional blindness online are both valid and reliable.

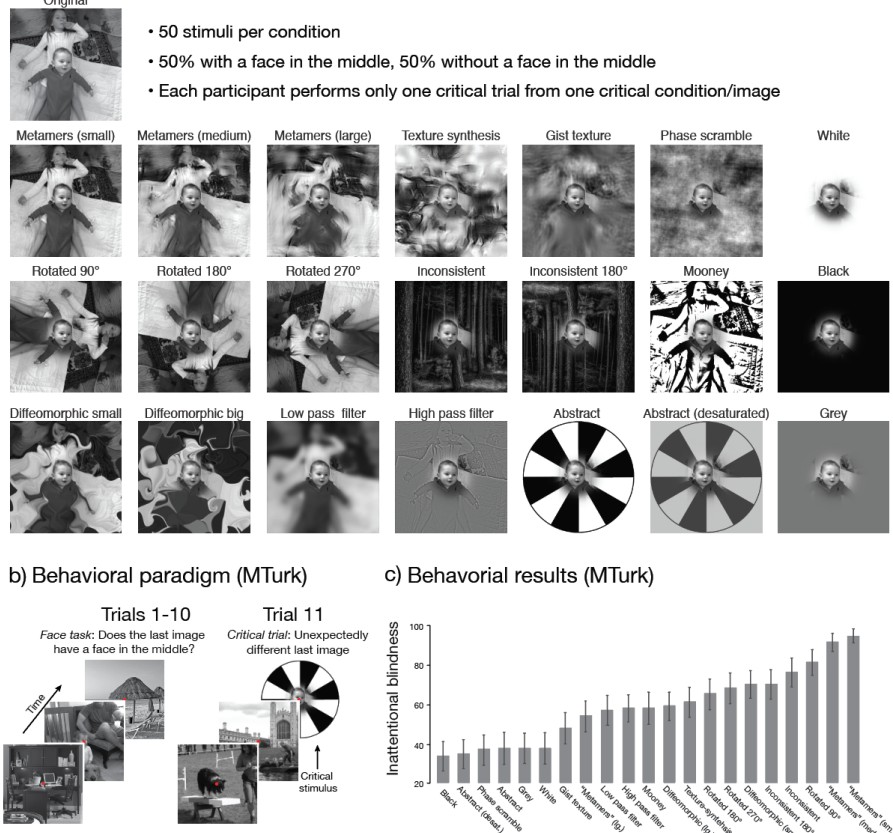

Figure 1: a) Stimuli. Examples from each experimental condition. An original image is shown on the top left, with an example of that image then being altered in each of the 21 different experimental condition. b) Visualization of the trial procedures for the behavioral experiment. Participants performed 10 trials where they simply said if the last image in the stream did or did not contain a human face in the middle. Then, on trial 11, an unexpected critical stimulus was presented at the end of the trial and participants were immediately probed to determine whether or not they noticed the critical stimulus. b) Inattentional blindness rates for each condition in the behavioral experiment. The percentage of participants who failed to notice the critical stimulus is plotted on the vertical axis. Each bar corresponds to a different experimental condition. The error bars represent bootstrapped standard errors.

## 3.2 Modeling behavior with deep convolutional neural networks (dCNNs)

How can we unify the behavioral results from these drastically different experimental conditions to form an overall understanding of perceptual awareness? To answer this question, we built predictive models of these behavioral findings, which we could then probe to identify the specific visual features that determine the bandwidth of perceptual experience. We screened several dCNN architectures to predict the observed behavioral data. This modeling approach is comprised of two parts: First, we measured the similarity between the features extracted for the original images and the altered images for each dCNN layer of a given network architecture. Then, we computed a linear mapping function between these similarity values and the behavioral measures (Figure 2a).

Which layers and features within a given network best predict the behavioral data? To answer this question, we calculated the correlations between the inattentional blindness rates and the cross-validated predicted inattentional blindness rates made by a given layer in each network architecture. This procedure was done with every layer of 7 architectures: AlexNet, VGG-16, VGG-19, ResNet-18, SqueezeNet, DarkNet19, and MobileNet. These architectures were chosen because they are somewhat similar in their depth relative to other networks (e.g., ResNet-50, GoogleNet, etc.), making it easier

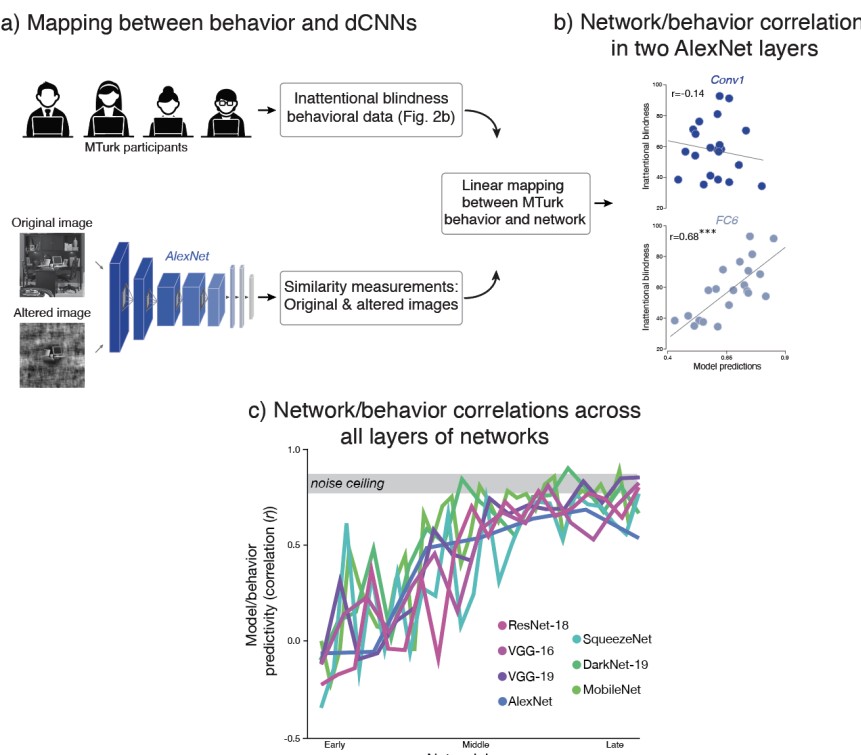

Figure 2: a) To create a predictive model, we computed a direct linear mapping between the behavioral data obtained on MTurk (Fig. 1c) and the similarity measurements between the original and altered images. b) With this model, we then calculated the cross-validated predicted inattentional blindness rates and correlated them with the observed inattentional blindness rates. c) The vertical axis represents the correlation between the observed behavioral results and a model's prediction on held out data (i.e., cross-validated). The grey bar represents the behavioral noise ceiling (see Methods). The horizontal axis represents the specific layer of a given network architecture. Each colored line corresponds to a given network.

to directly compare these networks to one another. The results from this analysis are plotted in Figure 2b. Across each network, we found that earlier layers could not predict the behavioral data. However, the later layers were able to predict the behavioral data, with many of these layers reaching the behavioral noise ceiling. Since numerous studies have shown that dCNNs such as these gradually build up abstractions across layers (i.e., from simple edges to textures to patterns to object parts, etc.), these results suggest that the extent to which an observer will notice the alterations to the periphery is directly related to the extent to which higher-level elements of a scene are preserved. As those higher-level elements are themselves altered, it increases the likelihood that a particular alteration will be noticed. Meanwhile, lower-level features can be altered without observers noticing, so long as these higher-level elements are aspects of an image are preserved.

An advantage of using computational models like dCNNs is that we can directly probe them to investigate the specific features that are linked with perceptual awareness. Here, we identified the specific features that drive the model's ability to predict behavior and visualized those features to get an intuitive understanding of what they represent. To identify the features with the most predictive power, we examined the regression model weights between the model features and the behavioral data. Then, we selected the 10 features with the highest weights and found that restricting the analyses to just these 10 features could predict the behavioral data to the noise ceiling (correlation with behavior r=0.83, P<0.00001).

What do these 10 features represent? To answer this question, we used a version of feature visualization to determine what attributes of scenes the top 10 features are representing. Here, we chose to directly observe the strength of the gradients. The advantage of this method is that it provides

maps that show the specific pixels that drive a given unit in a neural network the most. Specifically, we used a gradient attribution method called Guided Backpropagation ( Springenberg et al. (2015)). The results from this analysis results in what we call 'Attribution maps' and can be visualized for a few example stimuli in Figure 3. Overall, the attribution maps clearly focus on the outer contours of objects and scenes. For example, with an image of a canoe on a lake, the attribution maps highlight the outer contours of the canoe itself but do not focus on the texture properties of the water, which has no strong outer contour. Conversely, with a picture of a beach, in which the horizon and shoreline serve as clear contours, the attribution maps highlight both of these aspects of the scene. Indeed, after examining several examples, it becomes clear that the critical elements are the contours of a scene. In other words, the extent to which observers will notice alterations to an image appears to be linked to the extent to which those outer contours are preserved.

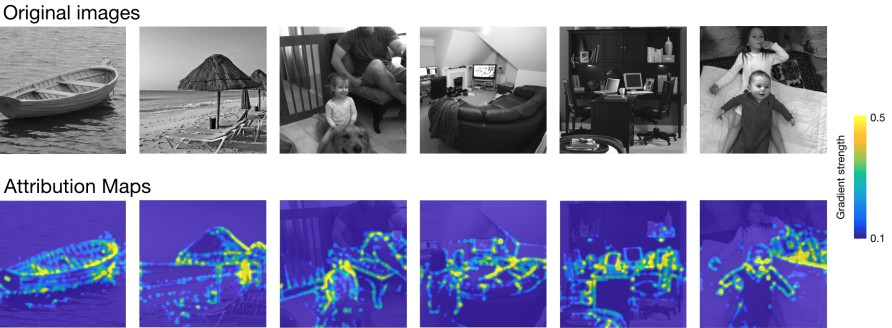

Figure 3: On the top row are six example original images. On the bottom row are visualizations from the Guided Backpropogation procedures. The gradient strength is plotted from blue to yellow

## 4 Conclusion

Here, we examined the bandwidth of visual awareness using an inattentional blindness paradigm with natural scenes. Specifically, we altered the periphery of natural images in a wide variety of manners and measured how often observers noticed those alterations. To gain insight as to which aspects of natural scenes drive these results, we screened several dCNN architectures to create a series of predictive models. Within each of these architectures, we found that later layers and higher-level features, but not earlier layers or lower-level features, could predict the behavioral results extremely well, reaching the noise ceiling in many cases. In addition, we used feature visualization techniques to directly examine the features that had the most predictive power. Overall, this analysis revealed that these particular features represented the contours of higher-level elements of a scene, such as those of complex objects (e.g., chairs, couches, people, etc.) and the largest contours of a scene (e.g., the horizon, the shoreline, etc.). Taken together, these results suggest that the extent to which observers will notice alterations in the periphery is dictated by the extent to which higher-level features are preserved in a given condition and suggest that perceptual awareness is limited by higher level aspects of a scene.

Overall, this set of results helps elucidate the contents of perceptual awareness by building predictive models of inattentional blindness in natural scenes. Moreover, this study also demonstrates how using deep learning techniques can help understand the bandwidth of perceptual awareness. Going forward, it will be important for researchers to continue developing these tools in order to fully explain the contents of human visual consciousness.

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
