# OpenReview forum: "The bandwidth of perceptual awareness is constrained by specific high-level visual features"
_NeurIPS.cc/2022/Workshop/SVRHM — SVRHM Oral_

### Official Review · Reviewer_GpXF · 2022-10-05
**Review for "The bandwidth of perceptual awareness is constrained by specific high-level visual features"**

**Rating:** 9
**Confidence:** 4

**Review:**

## Summary:

In the present paper the authors use an inattentional blindness paradigm to measure how often human participants notice various alterations of natural scenes. The authors develop a computational approach to describing the observed behavioral data using different CNNs. They find that features in higher layers but not early layers of the CNNs can predict the behavioral data highly accurately. These findings are taken as evidence that perceptual awareness is limited by high-level visual features.

## Pros:

- very clearly written and easy to follow
- the validation of the Mturk measurements with the in-lab measurements alleviates doubts about the validity of the Mturk approach
- CNN results hold true across various architectures
- Feature visualization supports the claims by providing a graphical description of the results

## Cons/Questions:

- the authors mention that deeper architectures such as ResNet-50 are not included since they are less comparable to the other networks. Based on the claims that the authors make, it could be expected that the effects also hold true in e.g. ResNet-50 (when considering depth relative to the other networks). Do these effects look different for deeper networks or was this simply an analysis choice of the authors ?
- the authors show the results of the feature visualization for some example images displaying that contours of e.g. prominent objects in the image are highlighted by the visualization method. I wonder if this holds true for all examples or if there are also examples for which this does not lead to interpretable results ? Is it possible to make a rough estimate of for how many of the examples this leads to interpretable/non-interpretable results ?

## Overall evaluation:

In sum, I really enjoyed reading the paper. It deals with an intriguing question, uses a well defined and validated experimental approach and develops a novel and highly interesting computational approach for describing the behavioral results. Further, the results clearly align with the conclusions by the authors. In my opinion this is a very valuable contribution to the workshop and probably one of the top submitted papers.

---

### Official Review · Reviewer_NwJ4 · 2022-10-10
**The importance of perceptual organisation**

**Rating:** 6
**Confidence:** 5

**Review:**

This paper provides evidence that "high-level" visual features (the borders of objects and other high-level structure in photographic images) are important in determining what information enters visual awareness in an inattentional blindness paradigm. Image manipulations that disrupt object borders and high-level structure are more noticable than manipulations that do not. The paper supports this via a large-scale online study and a laboratory study of an inattentional blindness task, followed by a linear readout strategy from dCNNs to predict the behavioural data.

## Strengths

- The paper is well written.
- The paper addresses an important problem.
- Validation with a lab sample is welcome.

## Weaknesses

- The experimental design is rather weak. Ideally the self-report questions used to determine the main outcome measure would be supplemented with a forced choice ("which trial contained an altered image?"), screening based on a free-text answer (i.e. get people to actually "say what it was"), or include a catch trial condition (in which there is no altered image in the last, critical trial) to measure the baseline rate of people answering "yes" to the three critical stimulus questions.

- The above concerns aside, the paper could usefully include a few sentences to define, distinguish and discuss "the bandwidth of perceptual awareness". Does an inattentional blindness paradigm (in which the participant is doing a separate distracting task) measure something qualitatively different from detection or discrimination tasks? Could one argue that inattentional blindness is not *perceptual* awareness but rather higher-level (ie. people may be more sensitive to disruptions if not performing a second task)? To what extent do results from detection or discrimination tasks corroborate or conflict with these findings?

- To somewhat answer my own last question: the overall conclusion seems to fit with other recent works in this domain, which are not cited. For example, the paper concludes "the extent to which observers will notice alterations to an image appears to be linked to the extent to which those outer contours are preserved" (line 129) and "the extent to which observers will notice alterations in the periphery is dictated by the extent to which higher-level features are preserved in a given condition and suggest that perceptual awareness is limited by higher level aspects of a scene" (142--145). These conclusions seem very similar to recent work by Wallis et al and Neri (see references below). Those authors argued for the importance of perceptual organization and/or object segmentation in determining the visibility of scene alterations. While this paper uses an inattentional blindness paradigm instead of e.g. detection paradigms, at a high level the conclusions seem similar. Please (1) cite the relevant prior work and/or (2) explain why the contributions here are different.

- In the dCNN analysis, to what extent does the main finding (Figure 2c) really support the conclusion above, vs some other feature of increasing depth (e.g. increasing receptive field sizes)? Figure 3 suggests it's not just about e.g. receptive field size increase, but showing the negative cases would strengthen the argument (i.e., show attribution maps for least predictive, high-layer features -- if they still appear as object contour heatmaps that would suggest that these outcomes could simply be a result of increasing RF size).

- In the feature selection analysis (lines 109-116), was crossvalidation or a hold-out sample used? Otherwise this analysis risks circularity (selecting highest-weighted features gives good prediction on the same data used for regression weight estimation).

## Minor

- Figure 1c: how many data points is each estimate based on?
- Figure 2c: Is there an estimate of uncertainty for these bars (e.g. different network instances trained from different weights)? The fact that there are negative correlations in two of the network early layers implies that these are quite noisy.
- insufficient anonymization (line 77 "our prior study")

## References

Neri, P. (2017). Object segmentation controls image reconstruction from natural scenes. PLOS Biology, 15(8), e1002611. https://doi.org/10.1371/journal.pbio.1002611

Wallis, T. S. A., Funke, C. M., Ecker, A. S., Gatys, L. A., Wichmann, F. A., & Bethge, M. (2019). Image content is more important than Bouma’s Law for scene metamers. ELife, 8, e42512. https://doi.org/10.7554/eLife.42512

---

### Official Review · Reviewer_2sDY · 2022-10-12
**A well written paper with interesting results!**

**Rating:** 8
**Confidence:** 5

**Review:**

Overall, I enjoyed reading this manuscript. It's well-written, clear, and easy to follow (except for a few typos, e.g. "blindness blindness").

I think the emphasis on "late layers" or "high-level features" needs reconsideration. First, the behaviour prediction for certain networks reaches the noise ceiling from the middle layers (Figure 2). Second, the interpretation of features represented in deep networks might not be as straightforward as we think. The feature space might be more convoluted across different levels of visual hierarchy.

I would suggest a follow-up experiment in which those pixels predicted by deep networks as influential (e.g. those visualised in Figure 3) are systematically perturbed, to ensure they impact intentional blindness.

The preregistration gives away the anonymity of the authors. The workshop organisers should decide how serious double-blindness is for them and make a decision accordingly.